# On the general relationship between plant height and aboveground biomass of vegetation stands in contrasted ecosystems

**Raphaël Proulx** *

Canada Research Chair in Ecological Integrity, Centre de recherche sur les interactions bassins versants-écosystèmes aquatiques, Université du Québec à Trois-Rivières, Trois-Rivières, Canada

* raphael.proulx@uqtr.ca

## Abstract

Ecological communities are unique assemblages of species that coexist in consequence of multi-causal processes that have proven hard to generalize. One possible exception are processes that control the biomass packing of vegetation stands; the amount of aboveground standing biomass expressed per unit volume. In this paper, I investigated the empirical and geometric underpinnings of biomass packing in terrestrial plant communities. I support that biomass packing in nature peaks around 1 kg m$^{-3}$ across contrasted contexts, ranging from grasslands to forest ecosystems. Using published experimental and long-term survey data, I show that expressing biomass per unit volume cancels the effects of air temperature, species richness and soil fertility on aboveground stocks, thus providing a general comparative measure of storage efficiency in plant communities.

## Introduction

Mass and stature are universal descriptors of living organisms that control fundamental processes like metabolic and dispersion rates [1, 2]. At the level of plant communities, stem height determines biological processes such as light interception, water evapotranspiration and seed dispersion [3]. As early as in 1902, Eichhorn discovered that volume production in a forest is a function of stand height only, irrespective of any difference in age [reviewed in 4]. A general relationship between mass and height among plant communities would thus have broad implications [5], especially for the rapid assessment of aboveground carbon stocks through remote sensing approaches. Furthermore, taking the ratio of mass, expressed per unit surface, over height for any plant community would provide a measure of biomass packing; that is, the amount of aboveground plant material that can be effectively stored per unit volume.

In the early 19[th] century, the Belgian statistician Quetelet reported that the mass of a human adult scales linearly to the square of its height, paving way to the body mass index (BMI; the ratio of mass to height squared) as a contemporary measure of physical condition [6]. BMI is used nowadays in routine exams to address mortality risk in large populations [7]. Similarly, fisheries have a long tradition of using the ratio of mass to length raised to the cube as a general measure of physical condition in fish populations [8]. More recently, Proulx and colleagues

**Data Availability Statement:** All data used in the present study are third party and accessible through the public domain, but restrictions apply to the availability of these data, which were used under license for the current study. We confirm

that these can be freely accessed by contacting the data owner directly. The aboveground biomass and aboveground height data that support the findings in this study are available from the Canada National Forest Inventory (https://nfi.nfis.org/data_and_ tools), the Cedar Creek Experiment (https://www. cedarcreek.umn.edu/research/data), and the Jena Experiment (https://figshare.com/collections/The_ Jena_Experiment_six_years_of_data_from_a_ grassland_biodiversity_experiment/3301988). Readers can access these data in the same manner as the authors. The author did not have any special access privileges to the data.

**Funding:** RP: Natural Sciences and Engineering Research Council (NSERC) Canada Research Chair (CRC) CRC-950-229255 https://www.chairs-chaires.gc.ca The funders had no role in study design, data collection and analysis, decision to publish, or preparation of the manuscript.

**Competing interests:** The authors have declared that no competing interests exist.

revisited data on hundredths of plant communities and showed that the ratio of mass over height cluster around 1 kg m$^{-3}$ and almost never exceeds 5 kg m$^{-3}$ across ecosystems ranging from biofuel crops to tropical forest stands [9]. The median and upper bounds reported by Proulx and colleagues are in good agreement with the handful of studies that directly measured the biomass packing of forest stands [10, 11], herbaceous communities [12], or submerged vegetation beds [13]. However, it is unclear whether external drivers, such as the species pool, climate, or soil conditions, affect the aboveground biomass packing of plant stands.

The objective of the present study was to evaluate the general relationship between dry aboveground biomass (AB) and stand height (H) across plant communities in contrasted ecosystems. More specifically, I tested if the biomass packing (the ratio of AB over H, expressed in kg m$^{-3}$) is independent of biotic and abiotic drivers. For this purpose, I retrieved AB and H from three long-term datasets on Western US prairies, Central Germany managed grasslands, and Canadian forests (Table 1) [14–16].

## Materials and methods

I retrieved the dry aboveground standing biomass and stand height of plant communities from three published long-term datasets: Western US prairies, Central Germany managed grasslands, and Canadian forests (Table 1). In each dataset, aboveground biomass was driven by nitrogen addition (Cedar Creek Experiment [14]), mean annual temperature (Canada National Forest Inventory [15]), or the richness of nitrogen-fixing legume species (Jena Experiment [16]).

The biomass of forest stands was estimated using protocols and models developed by the Canada's National Forest Inventory (CNFI). Aboveground biomass comprised stemwood, branch, bark and foliage modules for all saplings and trees in 813 forest stands across Canada. Stand height was reported as Lorey's height, which is the average height of all trees (> 9 cm dbh) weighted by their respective basal areas. Lorey's height was obtained by multiplying the tree height by its basal area, and then dividing the sum of this calculation by the total stand basal area. Another measure of stand height was also provided, which is the arithmetic average height of all trees. The two measures were highly correlated (r = 0.97) and yielded the same results. Forest stands with a biomass below 2 kg m$^{-2}$ were excluded because they were severely uncrowded (< 250 stems ha$^{-1}$), leaving 696 forest stands for the analyses. Mean annual temperature and stem density were provided by CNFI along with aboveground biomass and height data for each stand.

**Table 1. Summary of the three datasets used to compare the biomass packing of different plant communities and their responses to external drivers.** Aboveground biomass (AB) is the amount of dry standing plant tissue per unit area, while stand height (H) is the average height of canopy plants within the stand. Average photosynthetic height of canopy plants was used for herbaceous stands, whereas Lorey's height was used for forest stands.

| Dataset | No. stands | Sampling resolution | Mean aboveground biomass (± 1 SD) | Mean stand height (± 1SD) |
|---|---|---|---|---|
| Canada National Forest Inventory [a] | 696 | 400m$^2$ | 12.08 kg m$^{-2}$ (± 9.55) | 15.41 m (± 6.12) |
| | | (2000 to 2006) | | |
| Jena Experiment [b] | 640 | 0.4m$^2$ | 0.27 kg m$^{-2}$ (± 0.22) | 0.40 m (± 0.23) |
| | | (2003 to 2008) | | |
| Cedar Creek Experiment | 810 | 0.3m$^2$ | 0.33 kg m$^{-2}$ (± 0.19) | 0.42 m (± 0.15) |
| | | (1982 to 1986) | | |

[a] Forest stands with a biomass below 2 kg m$^{-2}$ were excluded because they were significantly uncrowded (< 250 stems ha$^{-1}$).

[b] Stands sown with only one plant species were excluded because these were heavily weeded and many did not grow a crowded cover (R. Proulx pers. obs.).

The aboveground biomass of herbaceous (prairie and managed grassland) stands was estimated through manual harvesting and drying of the vegetation. Stand height in these datasets was calculated as the average photosynthetic height (i.e., excluding flower structures) of randomly selected plants. In the Jena Experiment, stands sown with a single plant species were excluded because these were heavily weeded and many did not grow a crowded cover (R. Proulx pers. obs.). Nitrogen addition and plant species richness were design variables in the Cedar Creek Experiment and Jena Experiment, respectively.

Slope and intercept coefficients for the relationship between dry aboveground biomass and stand height were assessed through quantile regression on log-transformed data. Quantile regression is especially useful to describe how relationships behave at the boundary of the data envelope. Model coefficients and confidence intervals for $10^{th}$, $50^{th}$ and $90^{th}$ quantiles were calculated under the R environment using the rank method implemented in the quantreg package [17].

Determination coefficients ($R^2$) were estimated for the $50^{th}$ quantile (median) regression models as $1-[V_{res}/V_{dep}]$, where $V_{res}$ and $V_{dep}$ are variance terms for the model residual and dependent variable.

## Results and discussion

### Empirical underpinning

The biomass packing intercept (i.e., aboveground dry biomass at 1m height) of the relationship between AB and H across the pooled ecosystems and vegetation stands was 0.33, 0.68 and 1.19 kg m$^{-3}$ for $10^{th}$, $50^{th}$ and $90^{th}$ quantiles respectively, and the slope coefficient did not deviate from one (Fig 1). Overall, the height of vegetation stands explained 92% of the variation in AB. Biomass packing (the ratio of AB over H = BP) distributions were strikingly similar across the three ecosystems; with $95^{th}$ percentiles of 1.38, 1.41 and 1.40 kg m$^{-3}$ for forests, grasslands and prairies, respectively (S1 Fig).

Nitrogen addition and the number of legume species per plot explained, respectively, 30% and 17% of the variation in AB among experimental herbaceous communities. The mean annual temperature explained 14% of the variation in AB among plots of the Canada National Forest Inventory, which covers a broad range of climatic regions including deciduous, subalpine, boreal and coastal forests (both Pacific and Atlantic). When expressing biomass per unit volume, the percentage of explained variation in BP due to nitrogen addition, legume species richness, or annual temperature dropped to 6%, 2% and 7%, respectively (Fig 2).

### Geometric underpinning

Simple geometric principles tell us that the total aboveground biomass (TAB; kg) of a vegetation stand over a given area scales as follows:

$$TAB = N \cdot <CA> \cdot <BP> \cdot <H>, \tag{1}$$

where $< >$ are placeholders for the geometric mean of plant crown area (CA; m$^2$), biomass packing (BP; kg m$^{-3}$) and height (H; m) across the N individuals within the stand. If one considers a crowded stand in which each individual plant occupies a vital space proportional to its CA, so that N is inversely proportional to $<CA>$, then aboveground biomass (AB) is expressed per unit area as follows: AB = TAB·N$^{-1}$·$<CA>^{-1}$. Substituting AB in Eq 1 and log-transforming on both sides one gets: log AB = log $<BP>$ + log$<H>$. In support of this equation, reanalysis of the relationship between AB and $<H>$ for the large dataset compiled here revealed a common-group slope of ≈1 and median BP intercept (at 1m height) of ≈0.68 kg m$^-$

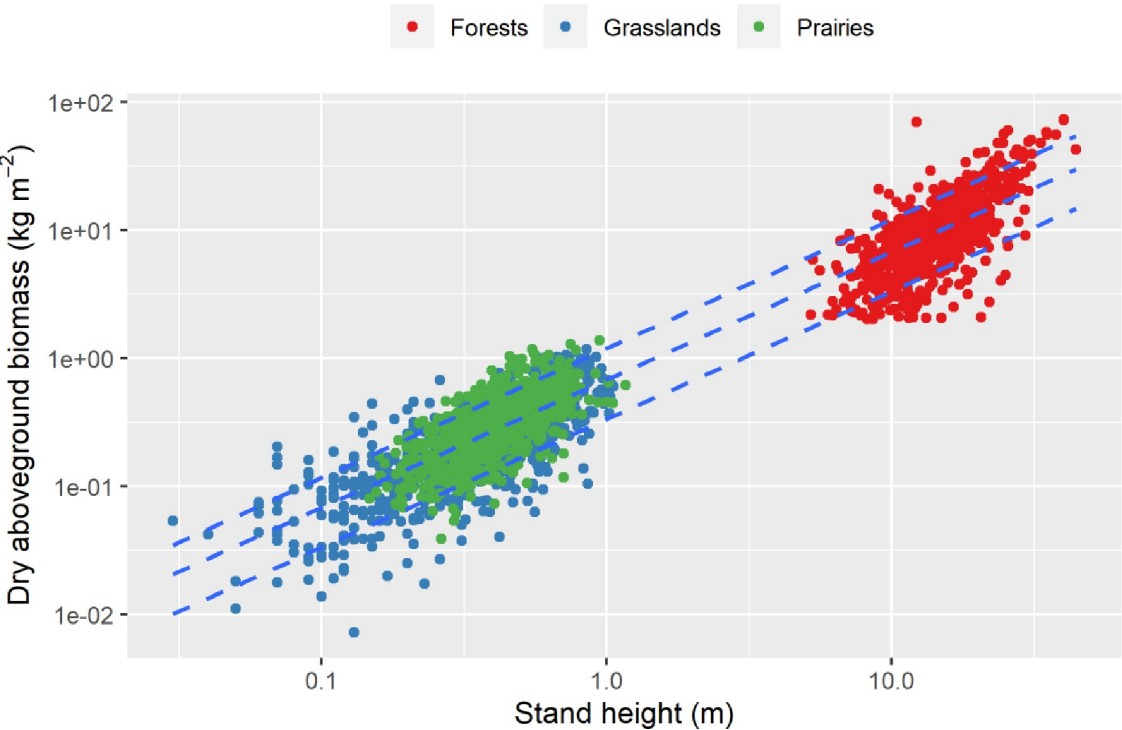

**Fig 1. Relationship between dry standing aboveground biomass (AB) and stand height (H) across 2,146 plant communities in three ecosystems: Canadian forests, Western US prairies and Central Germany managed grasslands (see Table 1).** Lines represent the fit of a power function of the form AB = <BP>·<H>$^b$, where $b$ and <BP> are the scaling exponent and the biomass packing intercept at 1m height, respectively. Model coefficients for different quantile regressions are as follows (95% confidence intervals in parentheses): Quantile 90th: $b$ = 1.001 (0.990; 1.021) and BP intercept = 1.191 kg m$^{-3}$ (1.167; 1.226). Quantile 50th: $b$ = 0.997 (0.987; 1.013) and BP intercept = 0.679 kg m$^{-3}$ (0.665; 0.696). Quantile 10th: $b$ = 0.999 (0.983; 1.027) and BP intercept = 0.334 (0.317; 0.346).

³ (Fig 1). The relationship in Eq (1) holds even if plant development is represented through elastic similarity rather than geometric similarity [18].

The above equation can be rescaled such that the individual aboveground biomass of an average stem (IAB = TAB·N$^{-1}$; kg) is proportional to its vital volume (VV = CA·H; m$^3$). Inserting IAB and VV in Eq 1 and taking the logarithm on both sides one gets: log IAB = log BP + log VV. In close agreement, reanalysis of existing data on 2,395 individual trees across the plant kingdom [19] revealed a log-log linear relationship between IAB and VV of slope ≈1 and median biomass packing intercept (at 1m$^3$ vital volume) of ≈0.42 kg m$^{-3}$ (S2 Fig).

## Discussion

This study highlights that the aboveground standing biomass of crowded vegetation stands is, for a large part, determined by plant height, which is in turn constrained by environmental drivers like soil nutrient and climate. A similarly general, strong and linear relationship between aboveground biomass and plant height was observed on an independent dataset of 75 vegetation stands [5]. I herein showed that communities sown with many nitrogen-fixing species, or fertilized in nitrogen, did not pack more biomass per unit volume, they mostly grew taller. Expressing biomass per unit volume thus cancels the effect of these drivers on aboveground biomass and provides a general surrogate measure of packing efficiency. Three independent datasets supported a high degree of overlap in the biomass packing distribution of plant communities across ecosystems; with 95th and 99th percentile packing values consistently

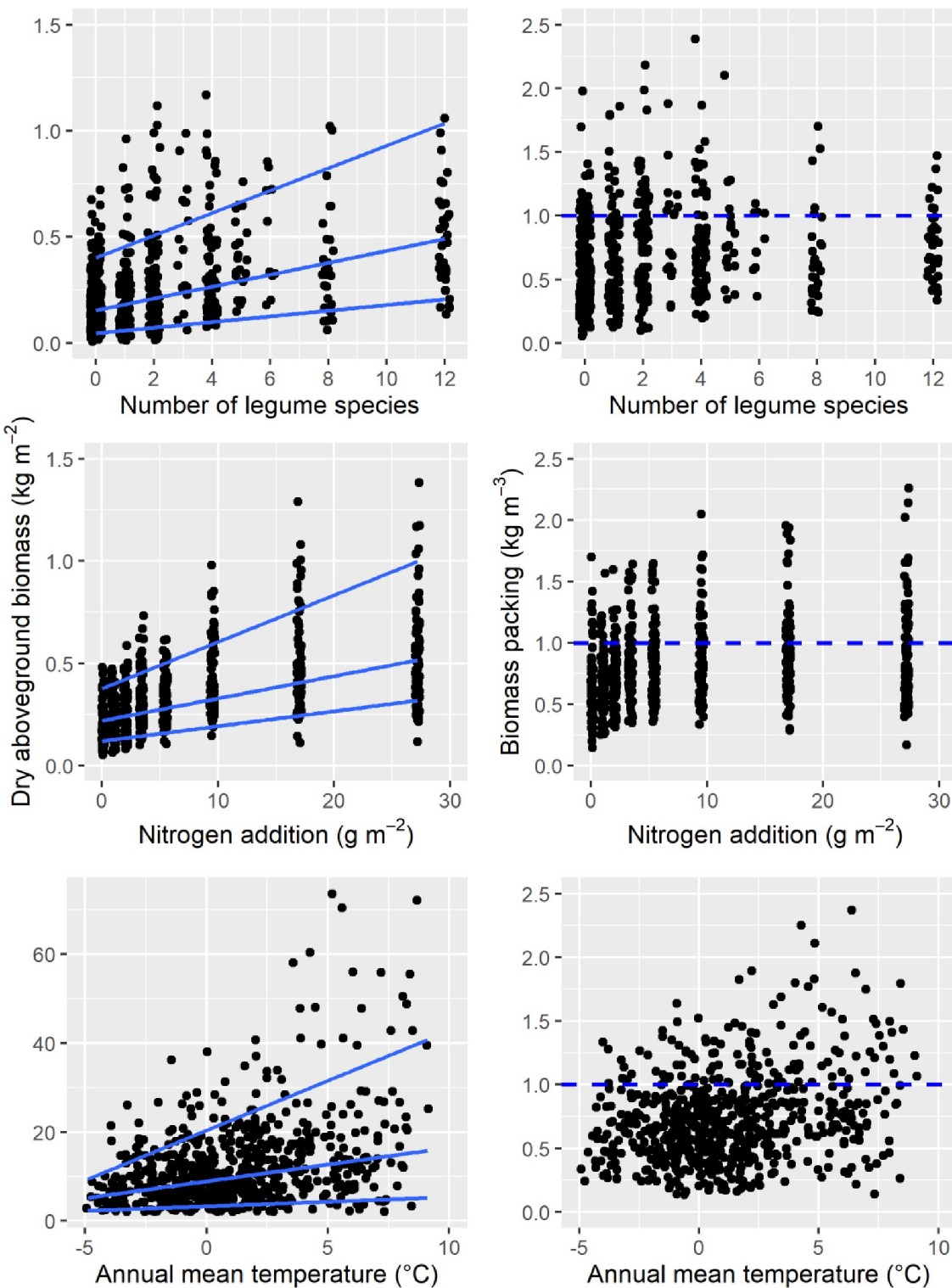

**Fig 2. Effect of legume species richness (top panels), nitrogen addition (middle panels), and annual mean temperature (bottom panels) on the dry aboveground biomass (left panels) and the biomass packing (right panels) of Central Germany managed grasslands, Western US prairies and Canadian forests.** Lines in left panels represent the 90th and 10th quantile regressions. Nitrogen addition, legume species richness, or annual temperature explained less than 7% of the variation in biomass packing through 50th quantile (median) regression. The horizontal dashed line indicates the 1 kg m$^{-3}$ reference value reported elsewhere [9].

falling close to 1.5 kg m$^{-3}$ and 3 kg m$^{-3}$, respectively. To put these results in a broader perspective, only photo-bioreactors designed for batch microalgae production typically approach a biomass-packing limit of ca. 5 kg m$^{-3}$ when there is plenty of available light and nutrient [20].

Plant geometry is the outcome of physical and physiological constraints imposed on stems and leaves. Using water displacement measures, I previously evaluated that the actual volume occupied by aerial tissues in wetland communities is typically close to 1% of the vital volume ($\approx 0.01$ m$^3$ m$^{-3}$) [9]. In comparison, stand height and aboveground stocks measured for 1,875 forest stands in Germany point to a more dramatic figure, with a meagre 0.1% of the volume occupied by vascular plant tissues ($\approx 0.001$ m$^3$ m$^{-3}$) [21]. Flipping these numbers around, one figures out that >99% of a plants' vital volume is air. The above observations suggest that the higher stem-over-leaf mass ratio of forests compared to herbaceous stands is balanced by a narrower vital volume used for the growth of aboveground modules; i.e., approximately 0.1% for trees and 1% for herbs [22]. Thus, although trees have denser stem tissues than herbs, the volume these tissues occupy is less dense by the same order of magnitude. As a result, the aboveground biomass of an individual plant stem scales isometrically with its vital volume (S2 Fig).

The mass index (i.e., biomass/length$^b$ ratio) is still one of the most widely used surrogate measures of physical condition in animal populations. Obviously, plants differ fundamentally from animals in many ways, including in that a fraction of their biomass is stored below-ground, or in "nonliving" heartwood structures. The biomass packing index presented here does not distinguish between aboveground modules (e.g., leaf vs. heartwood) and does not account for the belowground modules, and yet the metric remains generally applicable across broad geographic contexts. Why is that so? Related to the later, it is generally agreed that root and aboveground biomasses scale linearly across several orders of magnitude [22, 23]. In terms of nonliving heartwood, even though these structures do not metabolise *per se*, they contribute to energy storage and dissipation by supporting leaves and enhancing wood durability [24]. Root and heartwood modules coordinate towards an efficient storage of aboveground standing biomass in plant communities.

Differential mortality and growth are other sources of variation in the biomass packing of plant communities. Plants in vegetation stands may lose biomass when subjected to environmental stress or disturbances. For example, most deciduous forests leaf out each year as light and air temperatures decline, while prairies burn up or are grazed at recurring intervals. In the latter case, biomass packing may not change, or may even increase post-disturbances, because grazing or burning affect both stand height and aboveground biomass. In the former case, biomass packing should slightly decrease in winter because stand height does not vary much seasonally and most of the biomass is stored in wood. It should be noted however that the leaf mass fraction of short statured woody stands may reach up to 50% of the total biomass [22].

Whether residual spatial or temporal variation in biomass packing within an ecosystem results from stochastic events, local biotic interactions, or stress factors such as light or water deficit remains largely unexplored. The remaining variation in biomass packing observed among plant communities could relate to species-specific adaptations in resource use or storage. Thus, the upper limits of the distribution are probably more revealing of the constraints imposed on biomass packing than the lower limits, which may approach zero for severely uncrowded vegetation stand. Difference in protocols used to measure aboveground biomass and stand height across studies is another important source of variation for biomass packing. To attenuate this effect, the present study identified comprehensive data sources where stand height and aboveground biomass were i) measured using standard protocols and ii) influenced by experimentally manipulating biotic and abiotic environmental conditions.

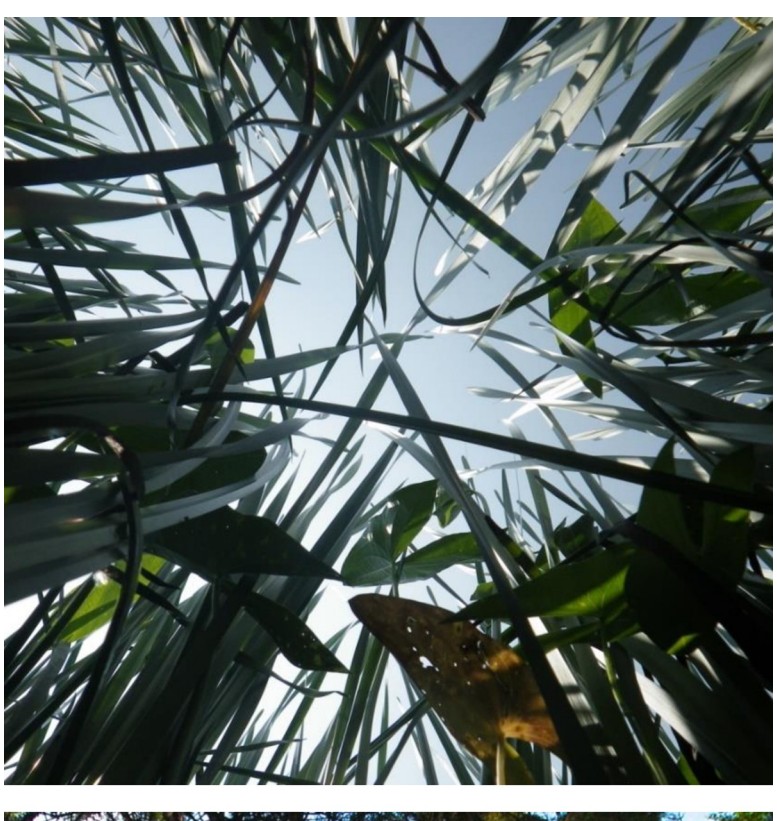

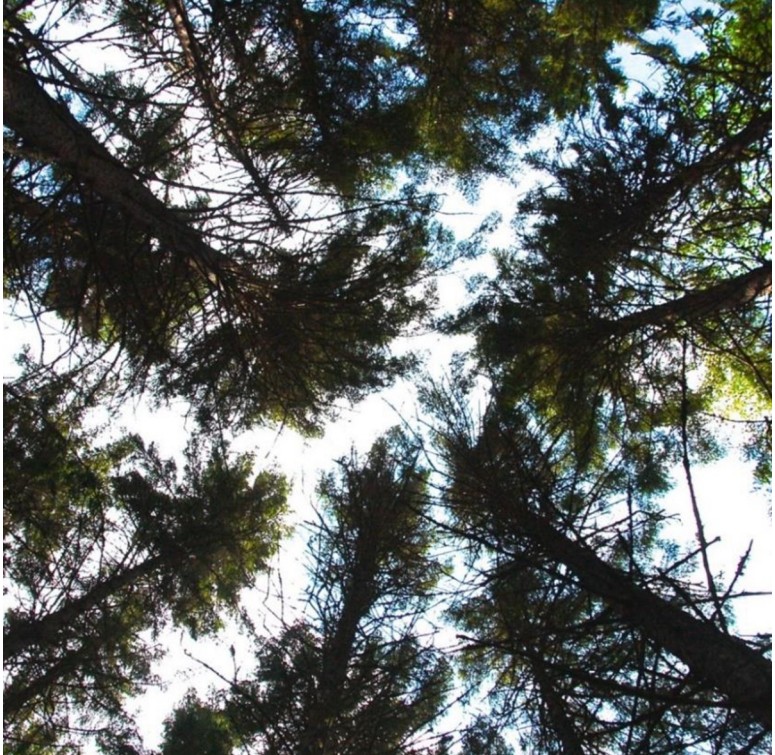

**Fig 3. Photographs of herbaceous and forest stands taken from underneath to illustrate their similarities in leaf cover (Photo credit R. Proulx and C. Martin.).**

## Conclusions

This study focusses on properties of plant communities that make them comparable rather than different (Fig 3). Species-area relationships and body size allometries are hallmarks of macro-ecology because they represent general, repeatable, patterns across scales. Biomass packing and mass-height relationships are stand-level patterns with similar potential for generalization. Although an individual tree could not be mistaken for an herb scaled up in size, the amount of standing biomass that can be packed per unit volume largely overlaps from small herbaceous to tall forest stands. Further understanding of this pattern could lay the foundation of a generally applicable mass index for plant communities.

## Supporting information

**S1 Fig. Biomass packing distribution of vegetation stands in three ecosystems: Canadian forests (National Forest Inventory), Central Germany managed grasslands (Jena Experiment) and Western US prairies (Cedar Creek Experiment).** Box hinges represent first and third quartiles (25th and 75th quantiles). 95th quantiles are 1.38, 1.41, 1.40 kg m$^{-3}$ for grasslands, prairies and forests, respectively.
(DOCX)

**S2 Fig. Relationship between individual aboveground dry biomass (IAB) and vital volume (VV = CA·H) for 2,395 individual trees across the plant kingdom [19].** The line is the fit of a power function of the form IAB = <BP>·VV$^b$, where b and <BP> are the scaling exponent and the biomass packing intercept at 1m$^3$ vital volume. Model coefficients for the 50$^{th}$ quantile (median) regression are as follows (95% confidences intervals in parentheses): $R^2$ = 0.86, b = 1.013 (0.988; 1.037) and BP intercept = 0.426 (0.389; 0.471) kg m$^{-3}$.
(DOCX)

## Acknowledgments

Thanks to the Cedar Creek Ecosystem Science Reserve, the Jena Biodiversity Experiment, the Canadian Forest Service, and all participating researchers for granting access to their data. Acknowledgements are made to everyone at the Centre de recherche sur les interactions bassins versants-écosystèmes aquatiques (RIVE) and Canada Research Chair in Ecological integrity (CRIE) for many stimulating discussions on biomass packing.

## Author Contributions

**Conceptualization:** Raphaël Proulx.

**Data curation:** Raphaël Proulx.

**Formal analysis:** Raphaël Proulx.

**Funding acquisition:** Raphaël Proulx.

**Investigation:** Raphaël Proulx.

**Methodology:** Raphaël Proulx.

**Project administration:** Raphaël Proulx.

**Writing – original draft:** Raphaël Proulx.

**Writing – review & editing:** Raphaël Proulx.

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
