## [Decision Letter · Decision Letter 0]

5 Feb 2021

PONE-D-20-37882

On the general relationship between plant height and aboveground biomass of vegetation stands in contrasted ecosystems

PLOS ONE

Dear Dr. Proulx,

Thank you for submitting your manuscript to PLOS ONE. After careful consideration, we feel that it has merit but does not fully meet PLOS ONE’s publication criteria as it currently stands. Therefore, we invite you to submit a revised version of the manuscript that addresses the points raised during the review process.

Specific Editor's comments:

As early as in 1902, Eichhorn discovered that stand volume production is a function of stand height only, i.e. the standing volume of a species at a specific stand height is the same across different sites (i.e. different environmental conditions). Later on, several authors, including prof. Assmann, confirmed the hypothesis. You may also read a review by Skovsgaard and Vanclay 2008 (https://doi.org/10.1093/forestry/cpm041). Although, the hypothesis of the biomass packing is indeed interesting, the hypothesis that the majority of the variation in aboveground standing biomas is explained by stand height is by far not knew.

Assmann E. Die Bedeutung des “erweiterten Eichhorn’schen Gesetzes” für die Konstruktion von Fichten-Ertragstafeln, Forstwiss. Centralbl., 1955, vol. 74 (pg. 321-330)

Eichhorn F. , Ertragstafeln für die Weistanne, 1902 Berlin, Verlag von Julius, Springer

J. P. Skovsgaard, J. K. Vanclay, Forest site productivity: a review of the evolution of dendrometric concepts for even-aged stands, Forestry: An International Journal of Forest Research, Volume 81, Issue 1, January 2008, Pages 13–31

Please, check if the structure of your manuscript, including reference style, follows criteria of PLOS ONE journal. For example, the section Material and Methods goes before Results, etc.

Please, clearly define the methods used - see comments of reviewers.

In addition, clarify the definition of stand height - mean, upper, top. If top, what is the percentage of trees to form top stand layer. Which variable (DBH, tree height or volume) was used to sort trees in the stand and to calculate the stand height (irrelevant if just the average height of trees was calculated)?

Any study published should be 100% reproducible, so please, provide all details of your methodology.

We look forward to receiving your revised manuscript.

Kind regards,

Michal Bosela, Ph.D.

Academic Editor

PLOS ONE

Journal Requirements:

2.We note that you have indicated that data from this study are available upon request. PLOS only allows data to be available upon request if there are legal or ethical restrictions on sharing data publicly. For more information on unacceptable data access restrictions, please see http://journals.plos.org/plosone/s/data-availability#loc-unacceptable-data-access-restrictions.

3.We note that the grant information you provided in the ‘Funding Information’ and ‘Financial Disclosure’ sections do not match.

Reviewers' comments:

Reviewer's Responses to Questions

**Comments to the Author**

1. Is the manuscript technically sound, and do the data support the conclusions?

Reviewer #1: Yes

Reviewer #2: Yes

2. Has the statistical analysis been performed appropriately and rigorously? 

Reviewer #1: Yes

Reviewer #2: Yes

3. Have the authors made all data underlying the findings in their manuscript fully available?

Reviewer #1: Yes

Reviewer #2: Yes

4. Is the manuscript presented in an intelligible fashion and written in standard English?

Reviewer #1: Yes

Reviewer #2: Yes

5. Review Comments to the Author

Reviewer #1: The manuscript under consideration shows that biomass packing maintains constant values across contrasted ecosystems being practically insignificant the effect of climate or site variables such as air temperature, species richness or soil fertility on aboveground stocks.

The paper sets interesting research questions. It should be of general interest to the readers of the journal. However, I consider that some minor corrections should be made before publication.

Abstract

Line 23. According to Fig. S1 biomass packing clusters around 0.6 kg m-3

Results

Lines 86-107. The theoretical underpinning is not a result of the analyzed data, but the equations that show the geometric relationships between the studied variables. I believe that these paragraphs should be placed in the introduction or in the methodology.

Fig. 1. It would be convenient to mark the points with a different color according to the database from which they originate (Western US prairies, Central Germany managed grasslands and Canadian forests).

Discusion

When the three different datasets are compared together, the slope of the AB-H relationship shown in Fig. 1 is close to 1. However, this figure also shows that, if each ecosystem is analyzed separately, this slope varies. For example, if we focus on the points belonging to the forest ecosystems, the slope is clearly greater than 1 and is apparently not linear. These discrepancies, between the general model and the results when we scale down and focus on a single ecosystem, should be discussed.

According to Fig. S1 the mean BP in the three types of ecosystems analyzed is very similar, close to 0.6. However in previous work (Proulx et al. 2015, DOI 10.7717/peerj.849) you analyze other ecosystems where the mean BP varies considerably. If ecosystems with very different BP values had been chosen for the study, would the same conclusions hold? Please discuss..

Methods

To see the analyses performed, it is necessary to go to the figure captions. Please specify in Methods section the statistical analyses that have been performed.

Please, show in a table the fitted equations with coefficients and p-values of each variable.

Specifies the software used and, if R software has been used, the specific package.

Lines 189-190: "Forest stands with a biomass below 1 kg m-2 were excluded". However, in Fig. 1 it seems that forest stands with less than 2 or 3 kg m-2 are missing.

Reviewer #2: The manuscript is presenting information to confirm those showed previously by Proulx et al (2015, PeerJ) about biomass packing, but now he was using an allometric approach.

The approach of testing the hypothesis of slope coefficient =1.00 for the relationship between aboveground biomass and height for different ecosystems (prairies, grasslands and forests) is supported by the difference between them and the high number of cases. But, from a rigorous point of view, the affirmation that the slope coefficient did not deviate from 1.00 (L74-75) need to be reconsidered as the CI were not including the 1.00 value. The same comment for the relationship between IAB and VV, from a rigorous statistic point of view the CI were not including the 1.00 value and in my opinion the sentence should be rewritten.

Moreover, I miss the inclusion of multilayer ecosystems as, for example, tropical ecosystems. In those ecosystems, trees shrub and herbaceous layer are existing together.

Figure 1 is showing the biomass-height relationship, but there is gap of data between 1 to 5/6 m height that could be cover by other ecosystem type (e.g., shrubs, or ecosystem with trees in a young development stage). It would be very suitable to present a continuous dataset for this relationship in order to improve the applicability of the results.

L155: Following Poorter et al. (2015, New Phytol), leaf mass fraction is an important biomass compartment. From their data, leaf mass in a tree about 100 kg could be between 7-8% of the aboveground biomass. So, I suggest to consider this sentence accordingly to the comment.

Minor comments:

L56-57 and L178-179: The order to present the different datasets are the opposite that showed in Table 1.

L75-76: the values that you were showing are not the same (at least they did not correspond each other) that showed in L308-309 (S1 Fig)

L137: I suggest: ‘of an individual plant stem scales near isometrically with its vital volume’

L145-146. It was also found that allometric scaling exponents were not always fixed and they vary with plant size (Poorter et al., 2015, New Phytol)

L153-154: I don’t think that the process of leaf out for deciduous trees when light and air temperature decrease should be considered as an example of environmental stress or disturbance.

L185: Following table 1 it should be ‘810 forest stands’

L190: The value that you presented ‘biomass below 1 kg m-2’ are not the same that showed in table 1

L280-283. The panels in figure 2 are not in the order that is presented in these lines. There is not panel for ‘age’, but you were showing a panel for ‘annual mean temperature’. Now, I have doubts about the order about the ecosystems as I would think that for number of legumes it should be for grasslands or prairies, N addition for grassland or prairies and annual mean temperature for forests. Please, you should check it deeply.

L285-286. In the Fig 2 (right panels) you were showing the biomass packing where the dashed line indicates the 1 kg m-3 packing. But in your analysis (line73) you presented that the biomass packing intercept was 0.62 kg m-3, and also in S1 Fig the mean values for the different ecosystems were different that 1.00 value. Was it a mistake and you changed to the scaling exponent? Please, it should be clarify.

6. PLOS authors have the option to publish the peer review history of their article (what does this mean?). If published, this will include your full peer review and any attached files.

Reviewer #1: No

Reviewer #2: No

---

## [Author Response · Author response to Decision Letter 0]

7 Apr 2021

Michal Bosela, Ph.D.

Academic Editor

PLOS ONE

Specific Editor's comments:

C1: As early as in 1902, Eichhorn discovered that stand volume production is a function of stand height only, i.e. the standing volume of a species at a specific stand height is the same across different sites (i.e. different environmental conditions). Later on, several authors, including prof. Assmann, confirmed the hypothesis. You may also read a review by Skovsgaard and Vanclay 2008 (https://doi.org/10.1093/forestry/cpm041). Although, the hypothesis of the biomass packing is indeed interesting, the hypothesis that the majority of the variation in aboveground standing biomas is explained by stand height is by far not knew.

Assmann E. Die Bedeutung des “erweiterten Eichhorn’schen Gesetzes” für die Konstruktion von Fichten-Ertragstafeln, Forstwiss. Centralbl., 1955, vol. 74.

Eichhorn F. , Ertragstafeln für die Weistanne, 1902 Berlin, Verlag von Julius, Springer

J. P. Skovsgaard, J. K. Vanclay, Forest site productivity: a review of the evolution of dendrometric concepts for even-aged stands, Forestry: An International Journal of Forest Research, Volume 81, Issue 1, January 2008, Pages 13–31.

R1: Great point. I inserted a sentence at line 33 to recognise Eichhorn’s legacy.

C2: Please, check if the structure of your manuscript, including reference style, follows criteria of PLOS ONE journal. For example, the section Material and Methods goes before Results, etc.

R2: The manuscript was originally transferred from PloS Biology. I have now followed PlosOne guidelines.

C3: Please, clearly define the methods used - see comments of reviewers. In addition, clarify the definition of stand height - mean, upper, top. If top, what is the percentage of trees to form top stand layer. Which variable (DBH, tree height or volume) was used to sort trees in the stand and to calculate the stand height (irrelevant if just the average height of trees was calculated)? Any study published should be 100% reproducible, so please, provide all details of your methodology.

R3: I agree. I have expanded the text at lines 78-106

Reviewers' comments:

Reviewer #1: The manuscript under consideration shows that biomass packing maintains constant values across contrasted ecosystems being practically insignificant the effect of climate or site variables such as air temperature, species richness or soil fertility on aboveground stocks. The paper sets interesting research questions. It should be of general interest to the readers of the journal. However, I consider that some minor corrections should be made before publication.

C4: Abstract Line 23. According to Fig. S1 biomass packing clusters around 0.6 kg m-3

R4: The sentence was modified at line 23 to: I support that biomass packing in nature peaks around 1 kg m-3 across contrasted contexts, ranging from grasslands to forest ecosystems.

C5: Results Lines 86-107. The theoretical underpinning is not a result of the analyzed data, but the equations that show the geometric relationships between the studied variables. I believe that these paragraphs should be placed in the introduction or in the methodology.

R5: I partly agree. This section provides simple equations that capture geometric relationships between stand’s aboveground compartments. However, I agree that the heading was perhaps misleading and changed it to Geometric underpinning.

C6: Fig. 1. It would be convenient to mark the points with a different color according to the database from which they originate (Western US prairies, Central Germany managed grasslands and Canadian forests).

R6: Done as requested.

C7: Discusion When the three different datasets are compared together, the slope of the AB-H relationship shown in Fig. 1 is close to 1. However, this figure also shows that, if each ecosystem is analyzed separately, this slope varies. For example, if we focus on the points belonging to the forest ecosystems, the slope is clearly greater than 1 and is apparently not linear. These discrepancies, between the general model and the results when we scale down and focus on a single ecosystem, should be discussed.

R7: I partly agree. The biomass packing index allows identifying plant stands that deviate from a common-group relationship across the plant kingdom. To better illustrate the generality of the relationships, I estimated relationships using quantile regression models. Quantile regression is especially useful to describe how relationships behave at the boundary of the data envelope. This allowed a robust estimation of model parameters and confidences intervals. Please see the response to comment C3 above.

C8: According to Fig. S1 the mean BP in the three types of ecosystems analyzed is very similar, close to 0.6. However in previous work (Proulx et al. 2015, DOI 10.7717/peerj.849) you analyze other ecosystems where the mean BP varies considerably. If ecosystems with very different BP values had been chosen for the study, would the same conclusions hold? Please discuss.

R8: I agree. This issue is acknowledged at line 206: Difference in protocols used to measure aboveground biomass and stand height across studies is another important source of variation for biomass packing. To attenuate this effect, the present study identified comprehensive data sources where stand height and aboveground biomass were i) measured using standard protocols and ii) influenced by experimentally manipulating biotic and abiotic environmental conditions.

C9: Methods: To see the analyses performed, it is necessary to go to the figure captions. Please specify in Methods section the statistical analyses that have been performed.

R9: Done as requested at lines 97-106.

C10: Please, show in a table the fitted equations with coefficients and p-values of each variable.

R10: The equation coefficients (and confidence intervals) are reported in the text and degrees of freedom (df) are found in Table 1. P-values are useless metrics here because the df are very high. I believe it would be a waste of space to devote a full table to these coefficients. However, I am inclined to include one if the editor deems it essential.

C11: Specifies the software used and, if R software has been used, the specific package.

R11: Done as requested at lines 97-106 and in the reference section.

C12: Lines 189-190: "Forest stands with a biomass below 1 kg m-2 were excluded". However, in Fig. 1 it seems that forest stands with less than 2 or 3 kg m-2 are missing.

R12: Thank you. I corrected the typo. The correct value is 2 km m-2.

Reviewer #2

C13: The manuscript is presenting information to confirm those showed previously by Proulx et al (2015, PeerJ) about biomass packing, but now he was using an allometric approach. The approach of testing the hypothesis of slope coefficient =1.00 for the relationship between aboveground biomass and height for different ecosystems (prairies, grasslands and forests) is supported by the difference between them and the high number of cases. But, from a rigorous point of view, the affirmation that the slope coefficient did not deviate from 1.00 (L74-75) need to be reconsidered as the CI were not including the 1.00 value. The same comment for the relationship between IAB and VV, from a rigorous statistic point of view the CI were not including the 1.00 value and in my opinion the sentence should be rewritten.

R13: Confidence intervals estimated using a more robust approach (quantile regression) overlap a slope of 1 in all cases. Please see the response to comment C7 above.

C14: Moreover, I miss the inclusion of multilayer ecosystems as, for example, tropical ecosystems. In those ecosystems, trees shrub and herbaceous layer are existing together.

Figure 1 is showing the biomass-height relationship, but there is gap of data between 1 to 5/6 m height that could be cover by other ecosystem type (e.g., shrubs, or ecosystem with trees in a young development stage). It would be very suitable to present a continuous dataset for this relationship in order to improve the applicability of the results.

R14: I understand. However, the present study targeted comprehensive data sources where stand height and aboveground biomass were i) measured using standard protocols and ii) influenced by experimentally manipulating biotic and abiotic environmental conditions. We plan to conduct a multi-ecosystem assessment of the biomass-packing index in a follow up paper.

C15: L155: Following Poorter et al. (2015, New Phytol), leaf mass fraction is an important biomass compartment. From their data, leaf mass in a tree about 100 kg could be between 7-8% of the aboveground biomass. So, I suggest to consider this sentence accordingly to the comment.

R15: I inserted a sentence at line 198 to provide more context: It should be noted however that the leaf mass fraction of short statured woody stands may reach up to 50% of the total biomass [20].

C16: Minor comments:

L56-57 and L178-179: The order to present the different datasets are the opposite that showed in Table 1.

L75-76: the values that you were showing are not the same (at least they did not correspond each other) that showed in L308-309 (S1 Fig)

L137: I suggest: ‘of an individual plant stem scales near isometrically with its vital volume’

L145-146. It was also found that allometric scaling exponents were not always fixed and they vary with plant size (Poorter et al., 2015, New Phytol)

L153-154: I don’t think that the process of leaf out for deciduous trees when light and air temperature decrease should be considered as an example of environmental stress or disturbance.

L185: Following table 1 it should be ‘810 forest stands’

L190: The value that you presented ‘biomass below 1 kg m-2’ are not the same that showed in table 1

L280-283. The panels in figure 2 are not in the order that is presented in these lines. There is not panel for ‘age’, but you were showing a panel for ‘annual mean temperature’. Now, I have doubts about the order about the ecosystems as I would think that for number of legumes it should be for grasslands or prairies, N addition for grassland or prairies and annual mean temperature for forests. Please, you should check it deeply.

L285-286. In the Fig 2 (right panels) you were showing the biomass packing where the dashed line indicates the 1 kg m-3 packing. But in your analysis (line73) you presented that the biomass packing intercept was 0.62 kg m-3, and also in S1 Fig the mean values for the different ecosystems were different that 1.00 value. Was it a mistake and you changed to the scaling exponent? Please, it should be clarify.

R16: All reported coefficients and numbers have been carefully checked.

---

## [Decision Letter · Decision Letter 1]

16 Apr 2021

PONE-D-20-37882R1

On the general relationship between plant height and aboveground biomass of vegetation stands in contrasted ecosystems

PLOS ONE

Dear Dr. Proulx,

Thank you for submitting your manuscript to PLOS ONE. After careful consideration, we feel that your manuscript can be acceptd for publication after minor revision. Please, consider a few final suggestions by Reviewer #1. I strongly encourage you to consider the suggestion of Reviewer #1 to add regression lines for each of the ecosystems separatelly in Figure 1 to see how the general trend applies for individual ecosystems. Beside this, your manuscript now reads well and can be accepted for publication.

We look forward to receiving your revised manuscript.

Kind regards,

Michal Bosela, Ph.D.

Academic Editor

PLOS ONE

Journal Requirements:

Reviewers' comments:

Reviewer's Responses to Questions

**Comments to the Author**

1. If the authors have adequately addressed your comments raised in a previous round of review and you feel that this manuscript is now acceptable for publication, you may indicate that here to bypass the “Comments to the Author” section, enter your conflict of interest statement in the “Confidential to Editor” section, and submit your "Accept" recommendation.

Reviewer #1: (No Response)

Reviewer #2: All comments have been addressed

2. Is the manuscript technically sound, and do the data support the conclusions?

Reviewer #1: Yes

Reviewer #2: Yes

3. Has the statistical analysis been performed appropriately and rigorously? 

Reviewer #1: Yes

Reviewer #2: Yes

4. Have the authors made all data underlying the findings in their manuscript fully available?

Reviewer #1: Yes

Reviewer #2: Yes

5. Is the manuscript presented in an intelligible fashion and written in standard English?

Reviewer #1: Yes

Reviewer #2: Yes

6. Review Comments to the Author

Reviewer #1: In general the authors have adequately addressed my comments. However, I would like the authors to analyze and discuss the pattern that BP follows within each ecosystem. Although it is not essential for the article, and if the editor does not consider it necessary this comment can be omitted, I think it would be very interesting to know if the general trend shown in Figure 1 holds true if each of the ecosystems is considered separately. In other words: with the data analyzed, it can be stated that BP remains constant between ecosystems but, does it also remain constant within each ecosystem? According to the graph it seems that it does not, especially in forests.

Reviewer #2: I appreciate the effort of the author to solve those points that could be less clear in the manuscript. You followed most of the suggestions from the Editor and reviewer comments and the document was enhanced. With the changes made, I think that the manuscript is ready for publishing now.

7. PLOS authors have the option to publish the peer review history of their article (what does this mean?). If published, this will include your full peer review and any attached files.

Reviewer #1: No

Reviewer #2: No

---

## [Author Response · Author response to Decision Letter 1]

5 May 2021

Dear Dr. Proulx,

Thank you for submitting your manuscript to PLOS ONE. After careful consideration, we feel that your manuscript can be acceptd for publication after minor revision. Please, consider a few final suggestions by Reviewer #1. 

C1: I strongly encourage you to consider the suggestion of Reviewer #1 to add regression lines for each of the ecosystems separatelly in Figure 1 to see how the general trend applies for individual ecosystems. Beside this, your manuscript now reads well and can be accepted for publication.

R1: I have added a S1 Table to the supplementary material section. The table reports slope and BP intercept coefficients (± 95% confidence intervals) for each dataset (ecosystem) separately. I have inserted text in the Result and Discussion sections.

At line 112: “Model coefficients estimated for each ecosystem separately showed more variation (S1 Table). Slopes coefficients were systematically higher for the forest dataset, but the confidence interval of the 10th quartile regression overlapped a slope of one.”

At line 212: “However, the approach relies on a few datasets only and cannot reveal systematic differences between ecosystems.”

I did not add regression lines for each ecosystem and quartile interval in Fig.1 because the display would be too crowded and difficult to interpret. While I agree that it is more transparent to report model coefficients for each ecosystem separately, it is worth emphasizing that between-ecosystem comparisons rely on only three datasets.

---

## [Editor Report · Decision Letter 2]

10 May 2021

On the general relationship between plant height and aboveground biomass of vegetation stands in contrasted ecosystems

PONE-D-20-37882R2

Dear Dr. Proulx,

We’re pleased to inform you that your manuscript has been judged scientifically suitable for publication and will be formally accepted for publication once it meets all outstanding technical requirements.

Kind regards,

Michal Bosela, Ph.D.

Academic Editor

PLOS ONE
---

## [Editor Report · Acceptance letter]

18 May 2021

PONE-D-20-37882R2 

On the general relationship between plant height and aboveground biomass of vegetation stands in contrasted ecosystems 

Dear Dr. Proulx:

I'm pleased to inform you that your manuscript has been deemed suitable for publication in PLOS ONE. Congratulations! Your manuscript is now with our production department. 

Kind regards, 

on behalf of

Dr. Michal Bosela 

Academic Editor

PLOS ONE